# Endocarp Development Study in Full Irrigated Olive Orchards and Impact on Fruit Features at Harvest

**DOI:** 10.3390/plants11243541

**Published:** 2022-12-15

**Authors:** Marta Sánchez-Piñero, María José Martín-Palomo, Alfonso Moriana, Mireia Corell, David Pérez-López

**Affiliations:** 1Departamento de Agronomía, ETSIA, Universidad de Sevilla, Crta. de Utrera Km 1, 41013 Seville, Spain; 2CSIC Associate Unit, Uso Sostenible del Suelo y el Agua en la Agricultura (Universidad de Sevilla-IRNAS), 41013 Seville, Spain; 3Departamento de Producción Agraria, CEIGRAM-Universidad Politécnica de Madrid, Av. Puerta de Hierro, 2, 28040 Madrid, Spain

**Keywords:** fruit growth, fruit dry weight, fruit size, pit-breaking pressure, pit hardening

## Abstract

Endocarp development in olive trees includes three periods: growth (Period I), massive sclerification (Period II) and maximum hardening (Period III). The two first are strongly related to yield and irrigation management. Period I was reported to coincide with mesocarp cell division and thus with final fruit size. Period II was considered to be the most drought-resistant phenological stage. However, little is known in olive trees about the length of these periods and their capacity for predicting fruit size at harvest. The aim of this work was to evaluate the length of both periods in different cultivars and different location of full irrigated orchards. We also aimed to study the fruit feature impact on harvest at the end of Period I. Data from full irrigated olive orchards of cv Cornicabra, Arbequina and Manzanilla in two different locations (Ciudad Real, Central Spain, and Seville, South Spain) were used. The pattern of pit-breaking pressure throughout the season was measured with fruit samples for several years (2006 to 2022). These data and climatic data were used to compare different estimation methods for the length of Period I and II of endocarp development. Then, fruit volume and dry weight at the end of Period I were used to estimate fruit features at harvest. Results suggest that the Period I length was less temperature- and cultivar-dependent than expected. The duration of this period was almost constant at around 49 days after full bloom. Thermal time was negatively correlated with fruit size at the end of Period I. On the contrary, a lineal thermal model presented the lowest variability when estimating the Period II length, which was also affected by the cultivar. The best fit between fruit dry weight and volume at Period I vs. harvest was unique for oil cultivars (Cornicabra and Arbequina), while cv Manzanilla presented a different relationship. A temperature increase in the future would not affect the Period I length but would reduce the fruit size at the end of this period and at harvest.

## 1. Introduction

Olive (*Olea europaea* L.) is a common rainfed fruit tree in the Mediterranean basin. However, from the 1990s, the increase in growing surface in this zone and around the world has been commonly achieved using irrigated orchards. Although this species is considered one of the most drought-resistant, water needs could still be very high [1]. In these new orchards, deficit irrigation managements were common because farmers’ available water was lower than the water needs. Regulated deficit irrigation works suggested that endocarp development was a critical period for water management. Endocarp hardening was reported as the most drought-resistant phenological period [2]. On the other hand, fruit set, coincident with endocarp growth [3], was determined to be very sensitive and related with reduction in fruit size of the current season and yield in the next season [4,5,6]. However, endocarp development was not commonly measured under field conditions.

The determination of the endocarp-hardening period during irrigation scheduling under field conditions and, also, in scientific works was not accurate. Commonly, fixed dates around July (North)/December (South hemisphere) were considered for different climatic conditions (e.g., [2,7,8]). Fruit longitudinal size could be used to estimate the beginning of the endocarp-hardening period [3]. This indicator has been used under field conditions for regulated deficit irrigation scheduling works with successful results in table [9,10,11] and oil olive orchards [12]. However, endocarp development was described as a dynamic process of three different periods [3], of which endocarp hardening is the second one. Ref. [13] reported that the sclerified area in the endocarp increased from full bloom, first gradually, then abruptly at the end of the endocarp expansion. These periods were also reported with the pit-breaking pressure pattern [3], with a slow increase in pressure (Period I) and fast increment (Period II). At the end of Period II, the endocarp reached maximum pit-breaking pressure, and the sclerification process was almost null [3].

Endocarp growth occurred only during the first period prior to the fast increase in pit-breaking pressure [3] at the beginning of Period II (pit-hardening period in irrigation literature). Period I of endocarp development was reported as coincident with the main part of mesocarp cell division activity [14]. This period was strongly related with yield, because mesocarp cell division established the final number of cells, and this was reported as the most important process on the determination of the final fruit size [14,15]. Then, in absence of water stress, fruit size at the end of Period I could be related with fruit size at harvest. Fruit size is a very important feature in table olive, but it also affects the decision of final use in cultivars either as oil or table (e.g., cv Hojiblanca).

This fruit growth pattern is well fitted to Mediterranean weather, but this could vary under changing climatic conditions. As far as we know, there were no scientific works which analyzed the influence of temperature in endocarp development. Endocarp hardening (Period II), the period of fruit growth less sensitive to water stress, occurred during the period of the year with higher water demand in the Mediterranean weather. This synchronization was reported to be closely related to temperature [16,17]. In the current context, global warming could change the best zone for olive production, which would be impacted through the effect on fruit and endocarp development.

The aim of this work was to study the length of the first two periods of endocarp development to model them and then use fruit weight or volume at the end of Period I for estimating harvest fruit size.

## 2. Results

### 2.1. Endocarp Hardening Model

A sigmoid equation was very well fitted for the pit-hardening pattern of all cultivars and locations considered (Table 1). The determination coefficients (R^2^) were higher than 0.7 in all cases, except in two of them, where variations occurred in the third period. Cv Manzanilla had significantly the hardest pit, with maximum values (a + d) of 167 MPa. Cv Cornicabra and cv Arbequina did not show significant differences for maximum pit hardening (130 MPa cv Cornicabra and 108 MPa cv Arbequina). The slope of pit hardening did not present significant differences between cultivars, with an average value of 0.14. This means an angle of 82° in the inflection point over the horizontal. The inflection point varied from 61 days after full bloom (DAFB) in cv Arbequina to 71 in cv Manzanilla with significant differences between them.

### 2.2. Period I of Endocarp Development

Different seasons and cultivars were compared in the characterization of the length of Period I of endocarp development (Table 2). Most data (8) were obtained in Ciudad Real (Central Spain), and only three were in Seville (South Spain). In Ciudad Real, full bloom was dated between the 22nd of May and 9th of June, while in Seville, it occurred in the second half of April (Table 3). Although there was a wide range of full bloom dates and cultivars, the length of Period I was very similar. The longest duration was cv Manzanilla in Seville during the season 2022 (53 days), while the shortest (46 days) was measured in Ciudad Real in different seasons and cultivars (Cornicabra, 2006; Arbequina, 2016 and 2017). The average length was around 49 days after full bloom, with a variation coefficient close to 5%. The three models which estimated thermal time in this period presented different results, but all of them were more variable than the length of days. The linear model was the most variable (cv = 13.7%). According to this model, this period needs between a 260° day (cv Manzanilla in Seville 2021) and a 470° day (cv Cornicabra in Ciudad Real 2012). This great variation (around 45%) was lower when the same cultivar was considered. Cornicabra changes from a 391 to 470° day, while Arbequina varied from a 371 to 442° day. Manzanilla, with only two data, presented the greatest variation. The exponential model showed a slight decrease in the variability between data (cv = 10.6%). This model estimated extreme values between 170° days (cv Manzanilla, Seville 2021) and 267° days (cv Cornicabra in Ciudad Real 2012), both cases are the same as in the previous model. However, the range of variations considered between these two extreme values was greater than 50%. When values of cv Manzanilla were not considered, the variability decreased from 226° days (cv Arbequina, Seville) to 267 (cv Cornicabra Ciudad Real, 2012): just a 15% of difference. The minimum variation coefficient was obtained with the cosine model (cv = 7.0%) close to variability when only the length in days was considered. In this latter model, the extreme values were different to the previous ones. The cosine model varied from 14,581° days (cv Cornicabra 2013) to 18,296° days (cv Arbequina, Seville). Such changes suppose a variation of 20%, which was the lowest when all cases were considered in thermal time models.

The thermal time during Period I was also compared with the pattern of fruit growth. Figure 1 and Figure 2 and Table 3 presented the relationship between relative dry weight and relative volume of fruits at the end of Period I vs. thermal time estimated with the three models (data in Table 2). Manzanilla data are not considered because only two seasons were available. Although relationships were not significant in all models and only a few data were available, in some of them, a tendency was found (Table 3). All data obtained negative correlations between relative volume and dry weight vs. thermal time (Figure 1 and Figure 2), which means more thermal time produced smaller fruits. The increase in thermal time decreased the weight and volume to 50–60% of maximum values. In both parameters, the cosine model was the worse relationship, while the exponential model was the best (Table 3).

### 2.3. Period II of Endocarp Development

The influence of thermal time and the length of Period II, massive hardening, varied between cultivars and locations (Table 4). The length of this period presented a greater variability than Period I (cv = 19% vs. 5.4%). The extreme values ranged from 26 days (cv Arbequina, Ciudad Real, 2017) to 50 days (cv Manzanilla, Seville 2021), which represents a 50% difference between both. When cultivars are considered separately, this variability decreased. Cornicabra ranged from 31 (2007) to 42 (2008) days, while Arbequina varied from 26 (Ciudad Real, 2017) to 38 days (Ciudad Real, 2008). Manzanilla also presented similar values in the two seasons considered, being longer overall than other varieties. The linear estimation model of thermal time presented the minimum variability between data (cv = 13.7%). Thermal time changes between 293° days (cv Arbequina, Ciudad Real, 2017) and 469°days (cv Manzanilla, Seville 2021): variations of 37%. When cultivars are considered, Arbequina tended to lower thermal time (from 293 to 368° days) than Cornicabra (from 307 to 419° days), and variability was also lower with changes between 20 and 25%. The exponential model presented a similar variability than the linear one (cv = 13.9%). Extreme values were obtained from the same data as a linear model (Manzanilla 2021 and Arbequina 2017) and with a similar variation (39%). This model also suggested slightly greater values in Cornicabra (from 177 to 229° days) than in Arbequina (from 167 to 206° days). Finally, the cosine model presented the greatest variability (cv = 22.4%). The pattern was the same as that in the previous model, with the extreme values in Arbequina 2017 (8220° days) and Manzanilla 2021 (17,806° days). This model also estimated that the thermal time was greater in Cornicabra (between 10,006 and 13,786° days) than in Arbequina (8220 to 12,143° days).

### 2.4. Prediction of Fruit Size

Relationships between fruit dry weight and volume at the end of the Period I of endocarp development vs. harvest presented a significant linear pattern (Figure 3 and Table 5). Although the volume and dry weight data of Arbequina were clearly lower than Cornicabra, the best fit included both data groups. Regression considered that each cultivar was worse than the one which included the pooled data. In weight data, Manzanilla also was included in the regression of Figure 3 and Table 5. On the contrary, Manzanilla volume data were out of the regression. The volume regression also presented the best fit when Cornicabra and Arbequina were considered together. Volume regression presented a more accurate fit than dry weight (Table 5), even when in the latter regression, only data of cv Cornicabra and Arbequina were considered (data not shown). The weight and volume increased around 2.5-folds since the end of Period I to harvest. When fruit load was included in a multivariable regression, the improvement was not significant in any of the regressions (data not shown). Fruit load is a factor that modifies the fruit size, so this result can be due to the use of several cultivars with different sizes that introduce a high scatter of data that is bigger than the effect of fruit load. More data per cultivar are necessary to quantify the fruit load effect. Maximum pit pressure at harvest (Table 1) was significantly linked to fruit dry weight at the end of Period I and at harvest (Figure 4). The best fit, in Figure 4, included the three cultivars considered in a linear regression. Each cultivar was grouped according to their weight, with the lowest values in cv Arbequina and the highest in cv Manzanilla.

The amount of data for cv Manzanilla was very small, just two seasons (Table 1, Table 2 and Table 3 and Figure 3). In order to evaluate the relationship between fruit volume at the end of Period I vs. harvest, published data were used [11,18]. Fruit volume at Period I was related to the volume at harvest (Figure 5a) and to the fruit size, which was estimated as the number of fruits per kilogram (Figure 5b). The accuracy of the regressions was similar to the ones obtained with Arbequina and Cornicabra (Table 6 vs. Table 5) but with a significantly greater slope. When a multivariable estimation was calculated using the fruit load, the improvements of the regression with volume data was low (Table 6). However, fruit load increased significantly the regression accuracy with fruit size (from 0.65 to 0.8, Table 6). The volume equation (Figure 5a), which did not consider fruit load (first equation in Table 6), included all measured data in cv Manzanilla (Figure 3b) in the prediction interval (95%).

## 3. Discussion

### 3.1. Endocarp Development

The length of the Period I of endocarp development was little affected by environmental conditions. The end of this period could be predicted accurately considering 49 days of length from full bloom and the average of all cultivars and seasons taken into account (Table 2). Environmental conditions, mainly temperature, would likely affect the length of this period, but the current approach suggested that the influence was low. Even the cultivar effect was very low, with variations shorter than 2 days when Arbequina and Cornicabra were compared. Moreover, the length of Period II (Table 4) was related to the DAFB of the inflection point (parameter c, Table 1; R^2^ = 0.6988 **, data not shown). This would support that the length of Period I was almost constant. Several authors with less seasons suggested a similar duration of Period I and did not consider the influence of temperature ([14] around 60 days; [3] around 50 days). The determination of the end of Period I of endocarp development is very important for the pattern of fruit growth. During Period I, the most important process in the endocarp and mesocarp was cell division [19]. Then, when this period finished, in mesocarp, the cell division decreased while the cell expansion continues, coinciding with Period II of endocarp development [3]. The main differences between cultivars in fruit size at harvest were related to the number of cells in the mesocarp at the end of Period I [14,20]. Endocarp was also reported to be the basis for the mesocarp growth [19], so a smaller endocarp, that grew during Period I, produced a smaller mesocarp and then a smaller fruit [5,19].

On the other hand, there was a significant influence of thermal time in the fruit growth during Period I (Figure 1, Table 3). Thermal time was negatively correlated with fruit volume and dry weight at the end of this period in all the models considered. This result suggests that also endocarp size would be affected because both are related [19]. Considering that current data correspond to full irrigated orchards, this implies that Period I occurred nowadays in a sub-optimal range of temperatures that could limit fruit development. Although olive trees are a fruit species adapted to warm climates, high temperatures would affect their growth capacity. These results mean that in the context of climatic changes, the increase in temperature would strongly reduce fruit size and, therefore, fruit yield, because Period I, with a constant length, would occur in a worse range of temperature for fruit growth. Olive field experiments simulating climatic changes with a 4 °C increase in temperature reported that the vegetative growth was reduced when the temperature reached values above 37 °C [21], as was fruit size [22]. However, [23] reported that the increase in temperature during a period similar to Period I of the current work did not affect the mesocarp dry weight, but [24] found a decrease in fruit dry weight for mean temperatures above 25 °C during the last period of fruit growth (about 110 days). These results could be related with the sort of experiment, where [23,24] only modify a branch while the rest of the tree retained the atmospheric temperature. Another possibility could be that cell expansion, the only cell process that occurs in the period in [24], is more sensitive to temperature than cell division. In both studies [23,24], oil concentration was negatively affected by mean growth temperatures above 20 °C.

The length of Period II of endocarp development was clearly affected by the temperature (Table 5). This period coincided with the most important part of sclerification (Table 1) which was different between cultivars (Table 1; Figure 4). This process had been related to a greater demand of reserves, which stops vegetative growth in mature trees [25]. The length of Period II varied from around 332 degree-days in Arbequina to 405 in Manzanilla (Table 4, Linear model) and was very closely related to maximum pit hardening (parameter a + d, Table 2; R^2^ = 0.7442 **, data not shown). Differences in maximum pit-breaking force between cultivars (Table 1) could be associated to a higher degree of sclerification or a greater endocarp with more sclerification cells. The relationship between maximum pit-breaking force (a + d parameters) and the fruit dry weight (Figure 4) suggested the second hypothesis. The increase in temperature in a context of climatic change would also affect Period II, but the final effect on yield would not be clear. This effect could be different between cultivars and would be related to the effect of temperature in the assimilation rate. Ref. [22] suggested that an increase in temperature would reduce the pulp–stone ratio and oil accumulation, but such an effect could be related to the decrease in fruit size produced in Period I [5,19]. The increase in air temperature during Period II in cv Araujo decreased the content of oil but not the dry weight in mesocarp [23].

### 3.2. Prediction of Fruit Size at Harvest

Fruit volume and dry weight at the end of Period I was an adequate predictor of fruit size at harvest (Figure 3 and Figure 5, Table 5 and Table 6). The estimation of size at harvest from data obtained early in the season is widely described in the literature for different fruit trees, included olive (among others, such as peach [26]; saskatoon [27]; olive [20,28]; apple [29]). In olive trees, the fruit size is strongly related to the cultivar and the number of cells in the fruit [28]. As most of the cell division occurred during Period I [14], the size at the end of this period would be linked to size at harvest. Water stress after this Period I would not affect fruit size when an adequate rehydration was performed [11,30]. This irrigation scheduling had been suggested for table olives, in which size was a very important feature, but oil olives could delay or even miss a complete rehydration [31]. Moreover, [11] reported that deficit management for table olives could be limited when a great fruit load was presented, because fruit size would be too small even in full irrigated conditions. Therefore, the estimation of fruit size at harvest could support the decision of water management during the season and even the final destination of yield if the fruit size expected was too small.

Fruit volume estimations suggested that table and oil cultivars could present different patterns of fruit development. In the current work, two oil cultivars (Arbequina and Cornicabra) had a common estimation, while cv Manzanilla, a table cultivar, was different (Figure 3 and Figure 5; Table 5 and Table 6). Cultivar Manzanilla presented greater volume at harvest and greater growth velocity throughout the season (Figure 3 and Figure 5). Fruit size is very important in table cultivars but so is the mesocarp vs. endocarp relationship [32]. This latter feature is greater in cv Manzanilla than in cv Cornicabra and Arbequina [33], and it could be related to the final fruit size [19].

## 4. Materials and Methods

### 4.1. Sites Description

Data were obtained from three different cultivars: Cornicabra (oil), Arbequina (oil) and Manzanilla (table), throughout several seasons (from 2006 to 2022). Trees were grown in different orchards located in Ciudad Real (central Spain) and Seville (south Spain). Both locations are around 350 km away and have different conditions for olive development (Table 7). Seville is in the main table olive-producing area in Spain, and climatic conditions are the optimum for olive growing. In this location, maximum temperatures were above 34 °C from May to September as an average for the period 2006–2021. On the contrary, winter was warm with minimum temperature slightly below 0 °C in January but higher than 6 °C from April to October. The rain pattern was the typical of a Mediterranean climate with a drought period in summer (from June to August). The average seasonal rain was 497 mm. On the contrary, climatic conditions in Ciudad Real could be limiting for olive development. Maximum temperatures were similar to Seville, but the period when they were above 34 °C was narrower (from June to September). Winter presented lower minimum temperatures for longer than Seville with values below −3 °C from November to March. The pattern of rains was similar in both locations, but Ciudad Real presented a lower seasonal rain than Seville with a total amount of 421 mm.

Eleven different seasons were used in the current work (Table 2):Five seasons for cultivar Cornicabra at “Entresierra” experimental farm near Ciudad Real (39° N, 3°56′ W; altitude 640 m).Three seasons for cv Arbequina at the same location.One season for cv Arbequina at “El Morillo” farm near Seville (37.5° N, 5.7° W; 102 m altitude).Two seasons for cv Manzanilla de Sevilla at “La Hampa” experimental farm near Seville (37° 17′ N, 6° 3′ W, 30 m altitude).

Orchards of cvs Cornicabra and Arbequina were mature (more than 10 years at the beginning of the experiment), and only the Manzanilla orchard was young (3 years, first yield in 2021 season). In the orchards located in Ciudad Real, trees were 7 m × 4.75 m apart, around 300 trees ha^−1^, while the two located in Seville were 4 m × 1.5 m apart, around 1667 trees ha^−1^. All data presented correspond to full irrigated treatments (100% crop evapotranspiration (ETc) throughout the irrigation period. Details about these orchards were published by [34] (Cornicabra), [12] (Arbequina, Seville) and [35] (Manzanilla).

### 4.2. Endocarp Development

The pattern of endocarp development was studied in these 11 seasons (Table 1). Full bloom was dated in all the seasons. Samples of 20 to 30 fruits were periodically collected during each season in 8 plots at Cornicabra orchard, 10 plots at Arbequina orchard in Ciudad Real, 4 plots at Arbequina orchard in Seville and 3 plots at Manzanilla orchard in Seville. Dry weight and fruit volume were measured weekly. Data from the end of period I and from harvesting were used for this work. The pit-breaking pressure in all fruits was measured according to [3] using a device similar to the one described in this latter work. In short, pressure applied by hand to a lever was transformed into the vertical movement of a probe terminating in a 2 mm diameter tip. Several capacity load cells of different pressure ranges were used. These load cells attached above the tip transformed the force applied to break the pit into an electrical signal. Pressure (MPa) is calculated as the force (N) divided by the area of the tip (m^2^). The electrical signal was sent to a data acquisition module (Model KUSB 318, Keithley Instruments Inc., Cleveland, OH, USA) connected by USB 2.0 to a computer for data processing. We also developed a specific application based on Visual Basic 6.0 to register the acquired data and to indicate on the screen the maximum pressure reached.

Figure 6 shows the expected pattern in full irrigated conditions for fruit dry weight and pit-breaking pressure according to [3]. The pattern of pit-breaking pressure was adjusted in each season using the sigmoid Equation [3]:(1)$$PBP=\frac{a}{1+e^{-b\left(DAFB-c\right)}}+d\;$$
where:

*PBP* was the pit-breaking pressure;

*a* is the range of PBP from the minimum to the maximum PBP value;

*b* is the slope coefficient at the inflection point;

*c* is the date at the inflection point;

*d* is the minimum PBP;

*DAFB* days after full bloom.

All seasons were adjusted using Equation (1) (Table 1). This pattern of endocarp hardening suggested three development periods (Figure 6 from [3]). The first period would be characterized by a slow hardening increase, which was modeled in the equation by a constant value (Equation (1), parameter “d”). The second period would be the massive hardening period with a fast increase in the pit-breaking pressure, as described in Equation (1) with the parameters “b” and “c”. Finally, the third period is when the increase in pit-breaking pressure is, again, very slow and modeled as a constant value (Equation (1), parameter “a + d”).

The beginning and the end of Period II (massive hardening period) were estimated using the same approach as [36] described for changing the linearity in pressure–volume curves. In short, the data of Periods I and III were obtained as the ones that maximize the lineal regression determination coefficient, using, at least, the first group (Period I) to the last (Period III) of 3 data. Each date was estimated as the intersection between these latter equations and the equation of fast increase in pit-breaking pressure (Period II). These dates allowed estimating the lengths of Period I and II.

The effect of temperature on the lengths of Periods I and II was evaluated with the estimation of thermal time. The thermal time for each season was estimated with three different models of degree-days (linear, exponential and cosine) using the half hourly data provided in Ciudad Real, La Puebla del Rio and Villanueva de Rio and Minas stations, by the Spanish Network of agri-climatic stations for irrigation [37] nearest to each orchard. The relationship between different biological processes and temperature was described as positive between two or three thresholds temperature, depending on the model considered. In the cosine model [38], three different thresholds were used: Lower Threshold Temperature (LTT), Optimum Temperature (OT) and Maximum Threshold Temperature (MTT). The response of any process speed to temperature would be an increase with temperature above an LTT. This rise stops when the temperature is higher than the OT, where it would be at a maximum. Above that temperature, the processes speed decreased until an MTT, where it would be zero. In the current work, LTT, OT and MTT were set to 4, 25 and 36 °C, respectively [39] for fruit trees. However, the most common models for thermal time simplified this with just two thresholds, LTT and MTT. The estimation of thermal time with these two thresholds could be modeled with linear and exponential equations [40] according to the expected response of the process to temperature. The most used model is the linear response, which is calculated as the mean daily temperature minus the LTT. In the current work, the result was divided by 48 to use half-hourly temperature values and obtain data that can be compared with other studies. The LTT threshold of these models (linear and exponential) was set according to the estimation of [16] for olive trees at 15 °C. The MTT value was set according to [40], who reported that olive trees that were acclimatized to high temperatures could maintain 70–80% of their maximum photosynthesis rate at 40 °C.

These three models of thermal time were also related to the relative dry and fruit volume at the end of Period I. The comparison of all data assumed very different values due to different cultivars; Cornicabra and Manzanilla were greater than Arbequina. In order to compare different years and cultivars, the relative fruit size of each season and cultivar was calculated.

### 4.3. Prediction of Fruit Size and Weight at Harvest

All data of fruit volume and dry weight at the end of Period I were related to the same parameters at harvest. The volume of data for cv Manzanilla was very scarce (just two years), and out of the obtained regressions, published and unpublished data of fruit volume were used to estimate the relationship between Period I and harvest for this cultivar. Published data [11,18] included three seasons of two Manzanilla olive orchards (2015 to 2017 and 2014 to 2016, respectively). Locations were: at Doña Ana farm, near Seville (37°25′ N, 5° 95′ W, 42 m altitude), 7 m × 4 m tree distance [11] and at La Hampa experimental farm, same as the above Manzanilla orchard but 7 m × 5 m tree distance [18]. Data from seasons 2020 and 2021 of the young Manzanilla orchard described above were also included. For all the seasons and orchards, 10 measurements of volume per plot in four [11], three [18] and six (young Manzanilla orchards) plots per orchard were measured periodically from full bloom until harvest. Data from the end of Period I were selected from the date nearest to the one expected, according to the results obtained in the first section of the current work.

### 4.4. Data Analysis

The lengths of Period I and II were compared using the thermal time models and days to minimize the variation coefficient. Linear regressions analyses were carried out to explore the relationships between different variables: volume and weight at the end of Period I and harvest, and also multivariable models with Statistic SX 8.0. Adjusted coefficients of determination (R^2^) were considered only in multi-variable models; otherwise, the determination coefficient was used. The prediction interval was estimated at 95% confidence for the best simple regression of published Manzanilla data to compare with current data.

## 5. Conclusions

Endocarp development was related to the thermal time in different ways. The length of Period I was constant (around 49 days after full bloom) and not clearly related to thermal time. However, fruit size at the end of this period was negatively correlated with thermal time and probably endocarp size. The period of hardening (Period II) was associated with the thermal time and was different for the various cultivars. In addition, fruit size at harvest was predicted with fruit size at the end of Period I with different relationships in oil and table cultivars. The constant length of Period I and current equations would support irrigation management and intended use of the yield. On the other hand, olive tree phenology closely fits the Mediterranean weather, which is characterized by warm winters but with temperatures low enough for flowering, bloom and early development (Period I) after a rainy spring to reach this time point without water stress. The fruit development takes place in a period very adapted to water stress (Period II), which occurs during the hottest and driest period of the summer. Current work suggests that climatic change would produce smaller fruits due to warmer temperatures during Period I, but the effect on Period II would not be clear in relation to yield.

## Figures and Tables

**Figure 1 plants-11-03541-f001:**
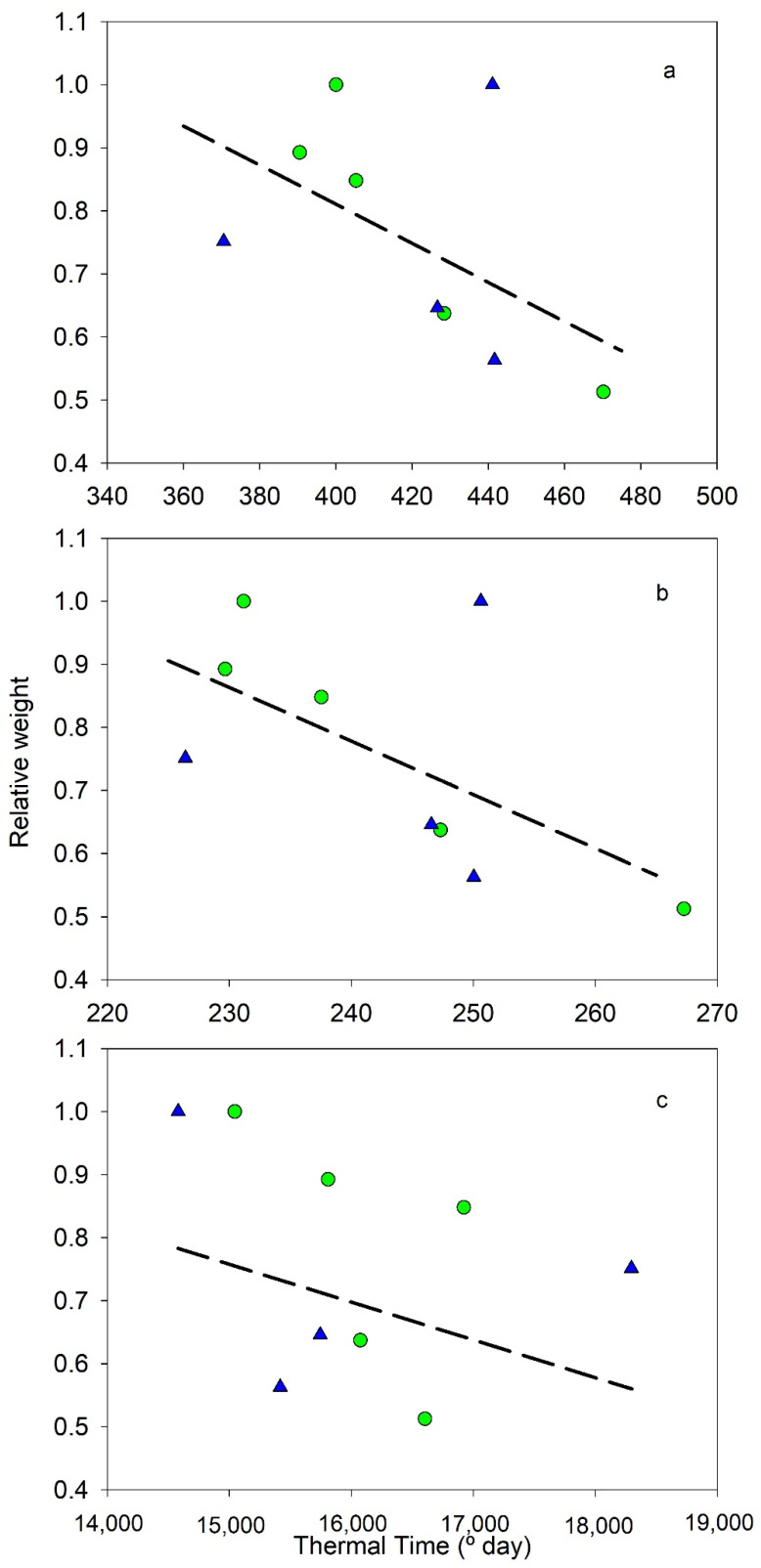
Relationship between relative fruit weight at the end of Period I of endocarp development and the three thermal time models (**a**) Lineal, (**b**) Exponential, (**c**) Cosine. Green circle, cv Cornicabra; blue triangle, cv Arbequina. Lines represent the best fit using all data. Each weight data are the average of the values obtained in each season.

**Figure 2 plants-11-03541-f002:**
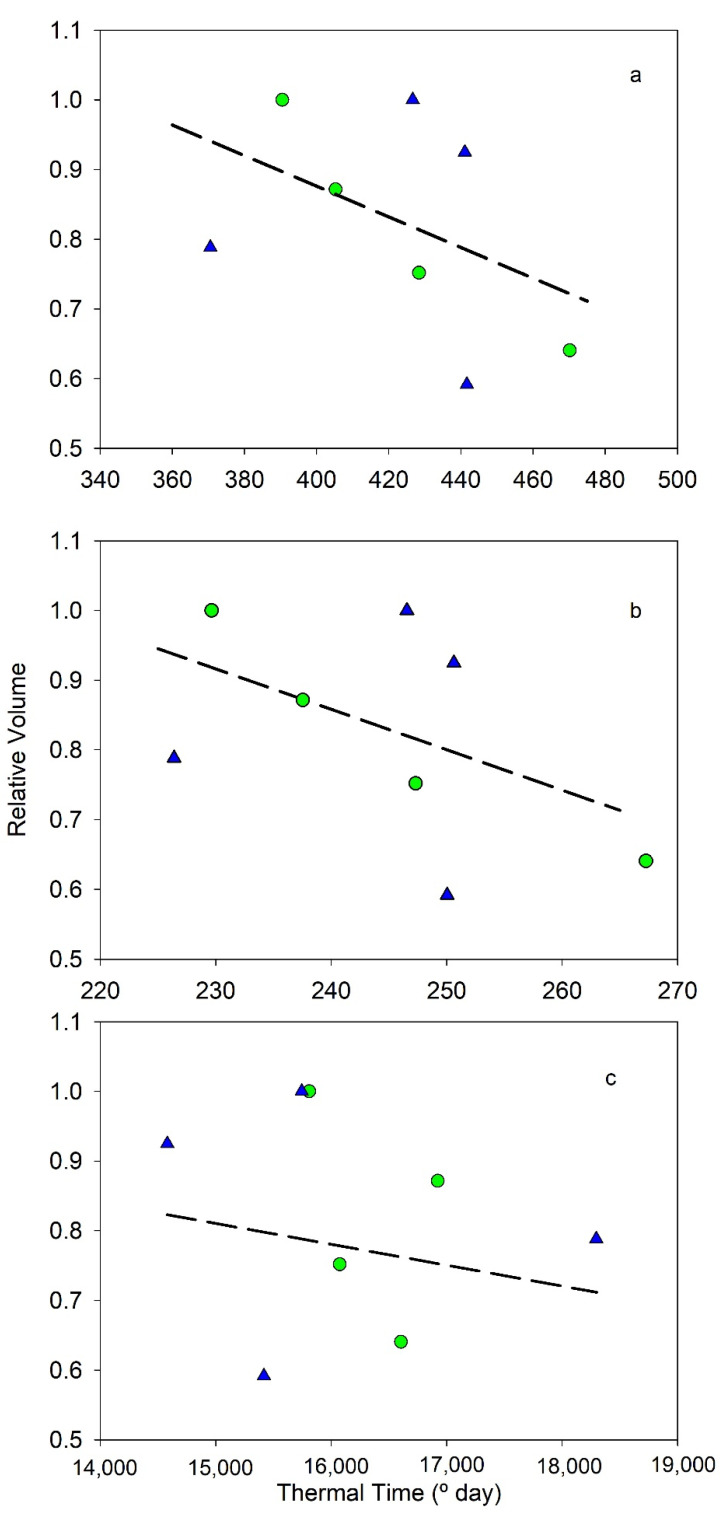
Relationship between relative fruit volume at the end of the Period I of endocarp development and the three thermal time models: (**a**) Lineal, (**b**) Exponential, and (**c**) Cosine. Green circle, cv Cornicabra; blue triangle, cv Arbequina. Lines represent the best fit using all data. Each weight data is the average of the values obtained in each season.

**Figure 3 plants-11-03541-f003:**
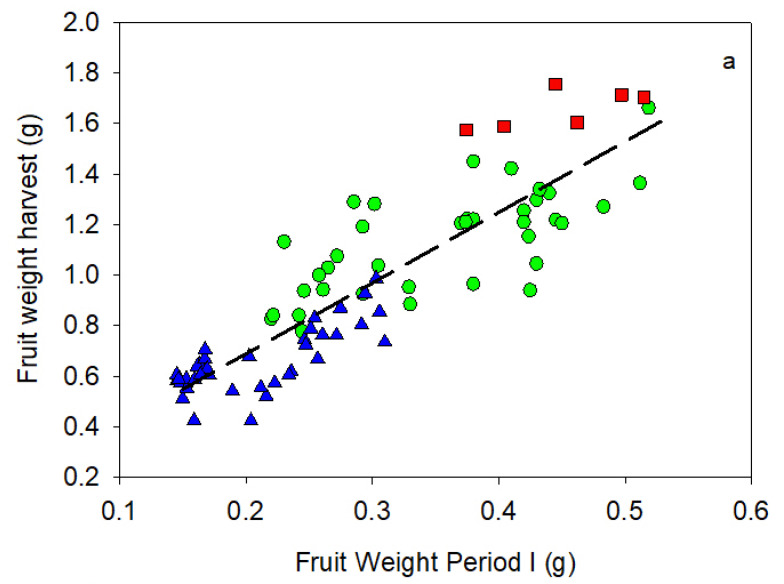
Relationship between fruit dry weight (**a**) and volume (**b**) at the end of Phase I of endocarp development vs. at harvest. Green circle, cv Cornicabra; Blue triangle, cv Arbequina; Red square, cv Manzanilla. Dashed lines represent the best fit with all data of cvs Cornicabra and Arbequina. Short dash lines represent the best fit obtained for cv Manzanilla with published data (see below). Each point is individual plot data of each season and cultivar.

**Figure 4 plants-11-03541-f004:**
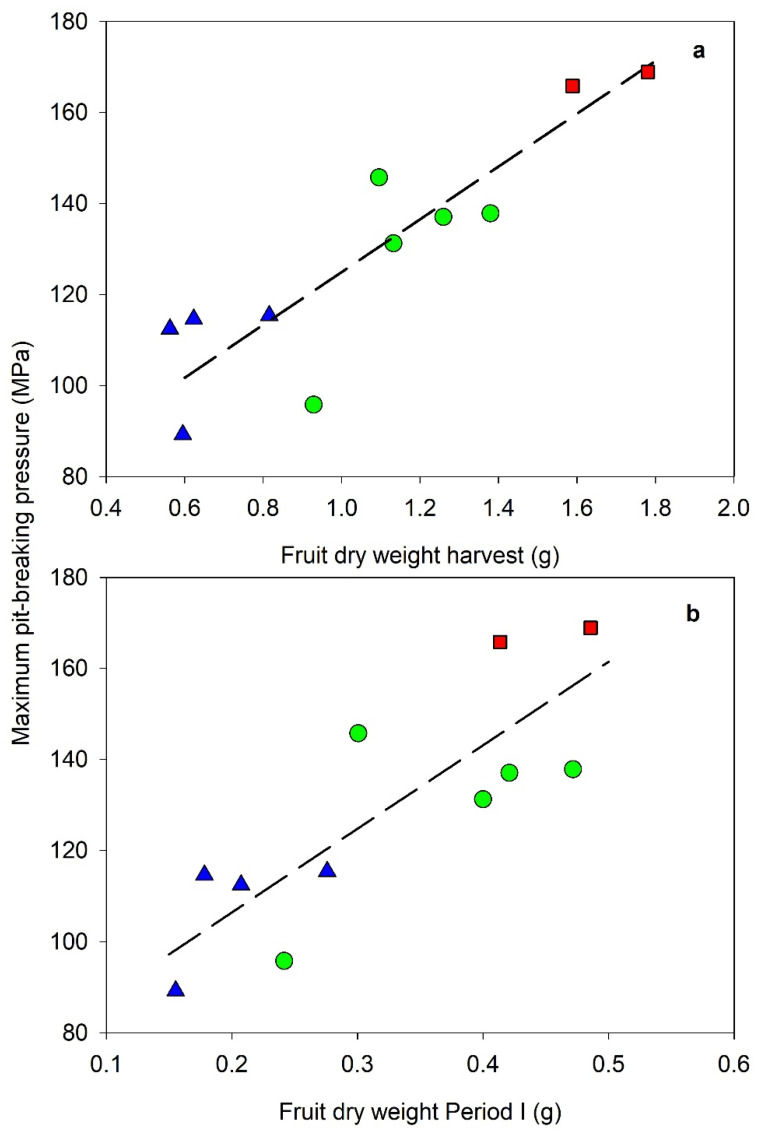
Relationship between fruit dry weight at harvest (**a**) and at the end of period I (**b**) vs. maximum pit-breaking pressure at harvest. Green circle, cv Cornicabra; Blue triangle, cv Arbequina; Red square, cv Manzanilla. Dashed lines represented the best fit with all data. (**a**) Pressure = 57.40 W + 67.43; N = 11; R^2^ = 0.78, Error = 12.7 MPa; MSE = 161.3. (**b**) Pressure = 178.53 W1 + 70.89; N = 11; R^2^ = 0.68, Error = 15.4 MPa; MSE = 236.6. Data are the average for each cultivar and season.

**Figure 5 plants-11-03541-f005:**
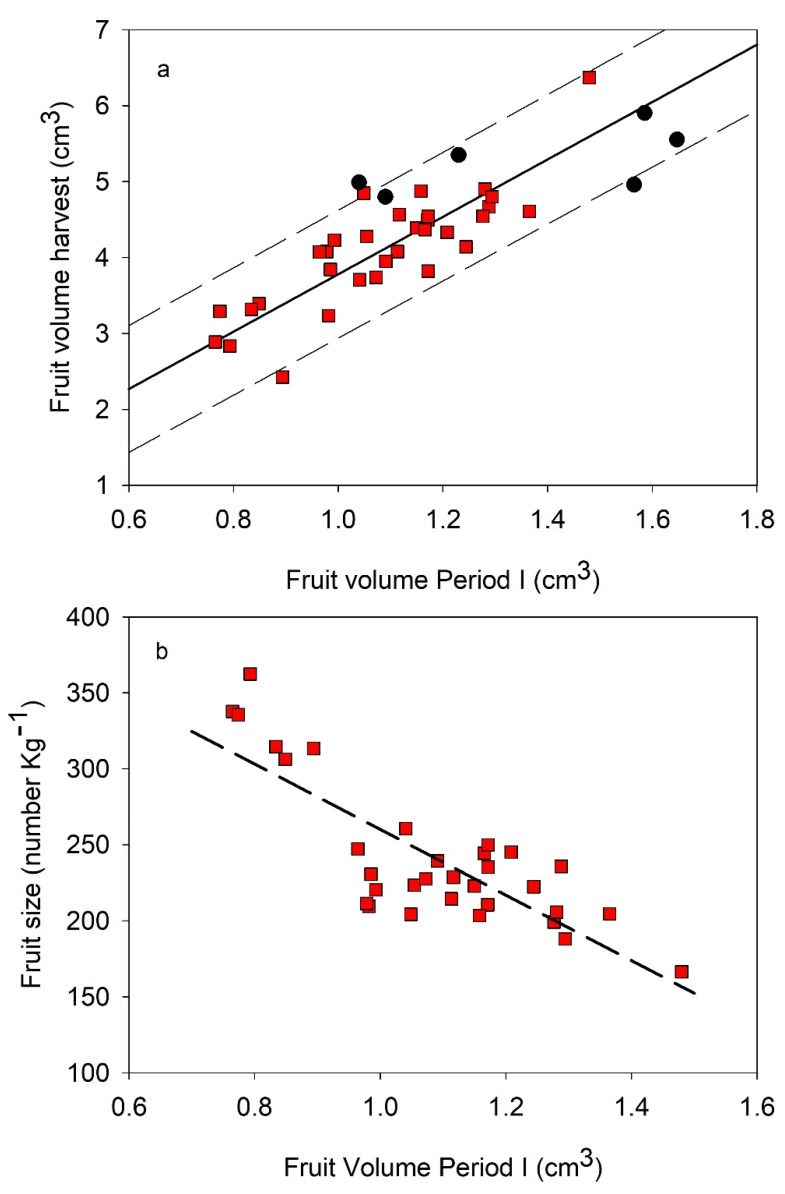
Relationship between fruit volume at Period I of endocarp development and fruit volume at harvest (**a**) and fruit size at harvest (**b**) of published Manzanilla experiments. Lines represent the best fit in both figures (equations at Table 6) and the prediction interval (95% confident) only (**a**). Black symbols at (**a**) are the Manzanilla data of Figure 3.

**Figure 6 plants-11-03541-f006:**
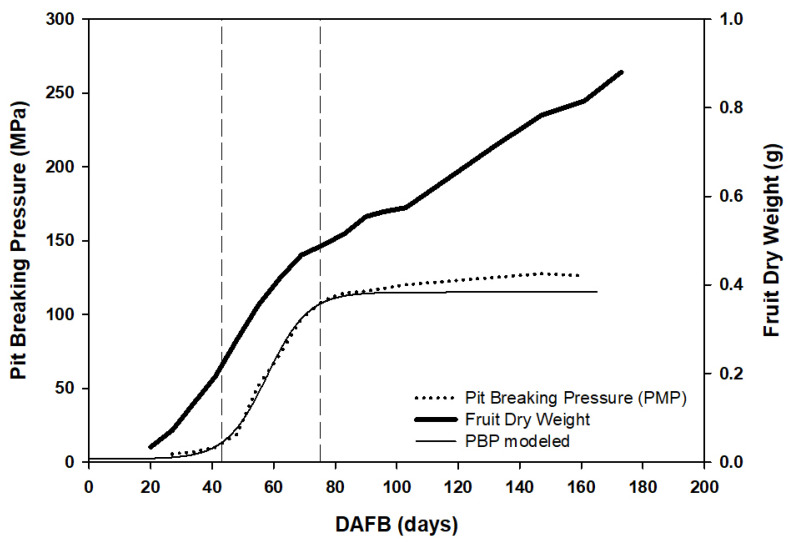
Scheme of seasonal pattern of fruit dry weight (bold line) and endocarp hardening (dotted line). Endocarp hardening was modeled by the equation $PBP=\frac{a}{1+e^{-b\left(DAFP-c\right)}}+d$ (solid line). Vertical lines define the three periods considering, according to the velocity of the hardening process.

**Table 1 plants-11-03541-t001:** Values of pit-breaking pressure (PBP) parameters and means and variability for the different varieties of measurement, according to the equation $PBP=\frac{a}{1+e^{-b\left(DAFP-c\right)}}+d$. Different letters in the same column mean significant differences between varieties.

Cultivar/Year	Location	a (MPa)	b (MPa Days^−1^)	c (days)	d (MPa)	a + d (MPa)	R^2^
Cornicabra 2006	Ciudad Real	134	0.1477	61	2.95	137	0.99
Cornicabra 2007	Ciudad Real	127	0.1626	64	4.53	131	0.99
Cornicabra 2008	Ciudad Real	151	0.1097	65	−4.90	146	0.94
Cornicabra 2012	Ciudad Real	136	0.1472	64	1.89	138	0.73
Cornicabra 2013	Ciudad Real	95	0.1192	64	1.19	96	0.55
Arbequina 2008	Ciudad Real	109	0.1413	64	3.45	112	0.98
Arbequina 2016	Ciudad Real	113	0.1500	58	2.09	115	0.56
Arbequina 2017	Ciudad Real	85	0.1832	57	4.74	89	0.95
Arbequina 2019	Seville	112	0.1434	66	2.56	114	0.88
Manzanilla 2021	Seville	156	0.1266	75	10.56	166	0.91
Manzanilla 2022	Seville	167	0.1102	68	1.49	168	0.95
Cornicabra		129 ab	0.14 ± 0.02	64 ab	1 ± 3.6	130 b	
Arbequina		105 b	0.16 ± 0.02	61 b	3 ± 1.2	108 b	
Manzanilla		162 a	0.12 ± 0.02	71 a	6 ± 7.5	167 a	

**Table 2 plants-11-03541-t002:** Characterization of Period I of endocarp development in different cultivars and locations: beginning (full bloom date), length (days), and thermal time (° days) according to three different models (lineal, exponential and cosine).

Cultivar	Location	Full Bloom	Length	Lineal	Exponential	Cosine
				LTT = 5 °C UPT = 40 °C	LTT = 5 °C UPT = 40 °C	LTT = 5 °C UPT = 40 °C OT = 25 °C
Cornicabra	Ciudad Real	25 May 2006	46	391	230	15,811
Cornicabra	Ciudad Real	7 June 2007	51	405	238	16,923
Cornicabra	Ciudad Real	7 June 2008	50	428	247	16,075
Arbequina	Ciudad Real	7 June 2008	47	400	231	15,044
Cornicabra	Ciudad Real	1 June 2012	51	470	267	16,604
Cornicabra	Ciudad Real	9 June 2013	47	441	251	14,581
Arbequina	Ciudad Real	8 June 2016	46	442	250	15,416
Arbequina	Ciudad Real	22 May 2017	46	427	247	15,746
Arbequina	Seville	25 April 2019	52	371	226	18,296
Manzanilla	Seville	17 April 2021	48	260	170	16,345
Manzanilla	Seville	27 April 2022	53	420	247	17,856
Cornicabra	Ciudad Real	4 June	49	427	246	15,999
Arbequina	Ciudad Real	2 June	46	423	243	15,402
Arbequina	Seville	25 April	52	371	226	18,296
Manzanilla	Seville	23 April	51	340	208	17,100
Average			48.8 ± 2.6	405 ± 55	237 ± 25	16,245 ± 1130
CV			5.4	13.7	10.6	7,0

Notes: LTT, Lower Threshold Temperature; UTT, Upper Threshold Temperature: OT, Optimum Temperature; CV, coefficient of variation (%).

**Table 3 plants-11-03541-t003:** Correlation coefficient and number of data in the relationships between relative weight and volume at the end of Period I of endocarp development with thermal time models.

	Correlation Coefficient (r)	R^2^	Mean Square Error	Number of Data
R Weight vs. Lineal Thermal time	−0.47	0.26	0.028	9
R Weight vs. Exp Thermal time	−0.53	0.36	0.025	9
R Weight vs. Cos Thermal time	−0.21	0.11	0.034	9
R Volume vs. Lineal Thermal time	−0.45	0.21	0.022	8
R Volume vs. Exp Thermal time	−0.49	0.24	0.021	8
R Volume vs. Cos Thermal time	−0.19	0.04	0.027	8
R Weight vs. R Volume	0.71	0.50	0.017	8

**Table 4 plants-11-03541-t004:** Characterization of Period II of pit-hardening period in different cultivars and locations: length (days), and thermal time (° days) according to three different models (lineal, exponential and cosine).

Cultivar	Locations	Length	Lineal	Exponential	Cosine
	Season		LTT = 5 °C UPT = 40 °C	LTT = 5 °C UPT = 40 °C	LTT = 5 °C UPT = 40 °C OT = 25 °C
Cornicabra	Ciudad Real 2006	35	419	229	10,575
Cornicabra	Ciudad Real 2007	31	307	177	10,456
Cornicabra	Ciudad Real 2008	42	383	218	13,786
Arbequina	Ciudad Real 2008	38	368	206	12,098
Cornicabra	Ciudad Real 2012	33	359	201	10,006
Cornicabra	Ciudad Real 2013	41	385	223	13,340
Arbequina	Ciudad Real 2016	31	352	195	10,445
Arbequina	Ciudad Real 2017	26	293	164	8220
Arbequina	Seville 2019	33	327	186	12,143
Manzanilla	Seville 2021	50	469	270	17,806
Manzanilla	Seville 2022	45	342	204	15,387
Cornicabra	Ciudad Real	37	371	210	11,633
Arbequina	Ciudad Real	32	338	188	10,254
Arbequina	Seville	33	327	186	12,143
Manzanilla	Seville	47	405	237	16,597
Average		36.9 ± 7.0	364 ± 50	207 ± 29	12,206 ± 2736
CV		19.0	13.7	13.9	22.4

Notes: LTT, Lower Threshold Temperature; UTT, Upper Threshold Temperature: OT, Optimum Temperature; CV, coefficient of variation.

**Table 5 plants-11-03541-t005:** Best fit of the relationships between dry weight and volume at Period 1 and harvest (Figure 4). The regression of dry weight included all data of the three cultivars. In the regression of volume, only data of cv Arbequina and Cornicabra were considered.

Best Fit	R^2^	Error	N	MSE
WH = 0.129 + 2.799 × W1	0.78 ***	0.164	81	0.027
VH = 2.482 × V1	0.87 ***	0.262	71	0.069

Notes: WH, weight at harvest; W1, weight at Phase I; VH, volume at harvest; V1, volume at Phase I; R^2^, coefficient of determination; Error, standard deviation; N, number of data; MSE, residual mean square error. *** indicates significate correlation at 0.001.

**Table 6 plants-11-03541-t006:** Best fits of the relationships between dry weight and volume at Period 1 vs. at harvest in Manzanilla experiments (Figure 5).

BEST Fit	R^2^	Error	N	MSE
VH = 3.781 × V1	0.71 ***	0.408	32	0.167
VH = 1.169 + 2.916 × V1 − 0.189 × FL	0.73 ***	0.391	32	0.153
FS = 475.6 − 215.6 × V1	0.65 ***	28.3	32	799.2
FS = 351.2 − 127.2 × V1 + 23.7 × FL	0.80 ***	21.0	32	440.4

Notes: VH, volume at harvest (cm^3^); V1, volume at Phase I (cm^3^); FL, fruit load (kg m^−3^); FS, fruit size (fruits kg^−1^); R^2^, coefficient of determination; Error, standard deviation; N, number of data; MSE, residual mean square error. *** indicates significate correlation at 0.001.

**Table 7 plants-11-03541-t007:** Comparison of the two locations where data of the experiments were obtained. Data are the average of monthly data in the period 2006 to 2021. Source of data: Climatic station of “La Rinconada” and “Ciudad Real” of the Spanish Network of agri-climatic stations for irrigation (SIAR, https://eportal.mapa.gob.es/websiar/SeleccionParametrosMap.aspx?dst=1, accessed on 15 October 2022).

	SEVILLE			CIUDAD REAL		
	T Max	T Min	Rain	T Max	T Min	Rain
January	19.88	−0.73	52	16.70	−6.12	37
February	22.49	0.44	52	19.27	−5.28	39
March	27.81	1.44	58	23.68	−3.51	46
April	30.37	5.95	62	26.31	−0.26	63
May	34.71	8.27	32	32.32	1.66	31
June	38.91	11.31	3	38.00	6.53	15
July	40.23	14.08	0	39.49	10.81	3
August	41.69	13.84	5	39.70	10.02	6
September	38.02	11.30	29	35.05	6.39	33
October	32.50	7.64	63	28.95	0.94	51
November	26.13	2.50	74	21.77	−3.62	52
December	20.77	0.11	67	16.70	−5.86	45

Note: T Max, average maximum temperature (°C); T Min, average minimum temperature (°C); Rain, average rain (mm).

## Data Availability

https://idus.us.es/handle/11441/32574 (accessed on 18 October 2022).

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
