# Peer review of "Endocarp Development Study in Full Irrigated Olive Orchards and Impact on Fruit Features at Harvest"

_plants, 2022, doi:10.3390/plants11243541_

Round 1

Reviewer 1 Report

I found the manuscript “Endocarp development study in full irrigated olive orchards and impact on fruit features at harvest” interesting, well perform, and well written. A general critic I have is regarding the emphasis on the cultivar differences in the responses. Because the cultivars were in different environments (locations and/or years), differences in response cannot be fully attributable to cv. Perhaps limiting the emphasis on cv to the discussion would be better. The other comment is related to the data presented in the figures. In some cases, each point represents the average value of the plots for the cultivar x environment, and in other cases, each point represents the individual plot for the cultivar x environment. Using the same criteria in all figures would help the reader to follow. Also, using the individual plot would probably better from a statistical point of view.   Other suggestions

1)      Table 1. I recommend adding the corresponding units to the parameters. This will help the reader to figure out which corresponds to forces related values or date related values.

2)     Line 142. Explain better. Are you talking about decreasing x (20?) Cd?

3)      Table 3. Add the level of significance or the sum of squares of the correlations to point out that none of them have good correlations with TT

4)      Lines 177-78. There is a confuse effect of cultivar, location and year, thus you cannot say that differences between varieties were more important than differences between location as even in the few cases that there were two varieties present at the same location, they were evaluated in different years, which represent a different environment

5)     Line 224. What do you mean by individual data? individual fruits? or the average of individual plots? 6)     Line 245. Include the references of the papers. If there are many, include a table as supplementary material

7)      Fig. 5b. The unit used in the figure for fruit size is kind of confusing. It would be better to calculate the individual fruit fresh weight from those data and use that variable in the y axis

8)      Line 273. In several annual crops the length of developmental periods is related to carbon acquisition and then to the intercepted radiation. Perhaps you can mention that (I have not reference to suggest in mind)

9)     Line 300. I recommend watching at Garcia Inza 2014 and 2016 works in which temperature was manipulated in range of temperatures during the whole fruit growth period or during seed growth and later mesocarp development. In the papers there are response functions of dry weight and oil to temperature.

10)  Table 7. Are this extreme values or mean values. If this are mean values, they look too high. How many days with temperature over 45 does it represent an average mean value of 41.69 C?. Similar comment applies to the -6 C in Ciudad Real

Line 434. Is this number correctly put here?
